# 3D Printing of Tooth Impressions Based on Multi-Detector Computed Tomography Images Combined with Beam Hardening Artifact Reduction in Metal Structures

**Yeon Park** [1,2] **and Seung-Man Yu** [2,*]

1   Department of Radiology, Seoul National University Hospital, Seoul 03080, Korea; amadeus29@hanmail.net
2   Department of Radiologic Science, College of Medical Sciences, Jeonju University, Jeonju 55069, Korea
*   Correspondence: ysm9993@jj.ac.kr; Tel.: +82-63-220-2382

**Abstract:** We investigated the role of metal artifact reduction by taking 3D print impressions using 3D data of Computed Tomography (CT) images based on the algorithm applied. We manufactured a phantom of a human mandible tooth made of gypsum and nickel alloy to measure the metal artifacts. CT images were obtained by changing the phantom tube voltage and tube current. The signal intensity of the image generated by the metal artifacts before and after the iterative metal artifact reduction algorithm (iMAR) was measured. A 3D printing process was performed after converting the images, before and after iMAR application, into STL files using InVesalius version 3.1.1 by selecting the conditions that minimized the effect of the artifact. Regarding metal artifacts, the Hounsfield unit (HU) value showed low as the tube voltage increased. The iMAR-applied images acquired under the same conditions showed a significantly lower HU. The artifacts, in the form of flashes, persisted in the 3D-printed product of the image not subjected to iMAR, but were largely removed in the 3D-printed product following iMAR application. In this study, the application of iMAR and data acquired using high tube voltage eliminated a significant portion of the metal artifacts, resulting in an impression shape that was consistent with the human body.

**Keywords:** 3D printing; metal artifact reduction; radiography

## 1. Introduction

During periodontal diseases such as tooth decay, the shape of the tooth is acquired using a dental impression. The treatment entails the insertion of a dental implant. An irreversible hydrocolloid material, known as alginate, is used to obtain the dental impression conventionally [1,2]. However, in this conventional method, since the alginate must be firmly maintained on the teeth, the patients may complain of discomfort during the acquisition method. The peculiar smell of alginate is not only repulsive to the patient but also affects the accuracy of the tooth shape during the patient's movement whilst taking the dental impression. In addition, areas in the tooth lacking firm bonding between alginate and the tooth during the extraction process can lead to errors in the tooth shape. Therefore, in order to overcome these limitations, and to alleviate patients' discomfort, digital data are being utilized to obtain impressions [3,4].

A 3D scanner does not generate the unique odor of alginate, and thus avoids patient discomfort. The 3D scanner acquires the 3D surface data of the teeth and the model of the tooth and, based on the 3D printout, generates an implant appropriate for the patient's periodontal disease [5,6]. The patient's information is processed digitally, obviating the need to store the patient's dental plaster model. The storage and management of the patient's dental information is easy and the optimal treatment method can be established via various simulations. The raw 3D data obtained with the scanner is converted into a stereophotography CAD software file (STL) and output to a 3D printer [7,8]. However, the accuracy of 3D scanners with digital advantages is limited in the oral cavity and by

changes in the scanner angle in the case of a patient with a small mouth or multiple structures in the oral cavity [9]. Recent advances in 3D printing technology using 3D medical imaging via computed tomography (CT) have been shown to overcome the disadvantages of 3D scanners and utilize the advantages of digital approaches [10,11].

CT images can accurately express the morphological data of a mandible, including the tooth, and retain the advantages of digital data, such as simulations before and after treatment. CT images not only provide accurate coordinates, but also facilitate the acquisition of data via X-rays externally, eliminating the need for a separate instrument for obtaining a patient's oral impressions [12]. Notably, since surface data can be extracted by controlling the degree of absorption attenuation of human tissue, information associated with the density of teeth can be selectively post-processed. Accurate diagnoses of the internal condition, in addition to the surface information of the teeth, enables customized treatment based on the patient's condition. CT images contribute to scalability of the treatment; however, beam hardening artifacts due to metal can occur in the case of patients with metal implants in their teeth. In fact, 3D printing is impossible if there is a metal artifact in the CT image, with almost no practical clinical, use until now. As X-rays pass through the metal structure, beam hardening artifacts, which are generated in the form of flashes due to the changed X-ray spectrum, adversely affect the pathological diagnosis. Several efforts have been made to reduce the artifacts. Recent efforts have been made to increase image uniformity by applying an iterative metal reduction algorithm (iMAR) [13–16]. However, taking impressions via 3D printing is not popular in the dental field to reduce such metal artifacts. The application of 3D printing by reducing metal artifacts has hardly been attempted in the dental field, until now. Therefore, the use of CT images in dental impressions could increase in the future if metal artifacts can be reduced.

First, in this study, we identified the optimal conditions for reducing metal artifacts by acquiring CT data under various conditions and applying iMAR to minimize the metal artifacts in CT images. In addition, the role of CT images acquired under optimal conditions in actual clinical practice was confirmed by taking 3D print impressions using the 3D data of the CT images before and after iMAR application.

## 2. Materials and Methods

### 2.1. CT Data Optimized for Metal Artifact Reduction and Phantom Composition

We used a self-made phantom in the form of a tooth made of plaster to measure the metal artifacts. The phantom contained all the shapes of human teeth and carried a nickel alloy dental prosthesis to express the dental prosthetic treatment virtually, which is often performed in patients with caries. Using this phantom, we measured the frequency of artifacts based on the effect of tube voltage change (kVp), the effect of tube current (mAs), and iMAR algorithm applied. All 64-channel multi-detector CT images were performed using a Somatom Syngo (Siemens, German) and CT scans were based on changes in five parameters. The position of the phantom was not changed according to the conditions. To measure the degree of artifact generation according to the change in tube voltage, the tube current was fixed at 50 mAs. The tube voltage was changed to 80, 100, or 120 kVp to acquire images in 3D using a helical scan method. In order to evaluate the tube current of artifacts, the tube voltage was fixed at 120 kVp and the tube current changed to 30, 50, and 80 mAs, and images were acquired in the same manner. Reconstructed images using the iMAR algorithm were obtained under altered conditions, and the degree of reduction in artifacts according to changes in tube voltage and tube current was compared and evaluated.

### 2.2. Application of iMAR Algorithm

The outline of the iMAR algorithm for CT images is presented in the schematic diagram in Figure 1.

In brief, an image that did not reduce metal artifacts from the original raw data was reconstructed. Next, only the metal data were extracted by designating the threshold Hounsfield unit (HU) value; hence, only the metal image was formed. Simultaneously, the

bone and soft tissue reconstruction images (*prior* images) were acquired as separate metal images, followed by conversion into sinograms to finally obtain a metal sinogram and a prior sinogram. Normalization was performed by dividing the original sinogram by the prior sinogram.

$$p^{corr} = P^{prior} M p^{norm} = p^{prior} M \frac{p}{p^{prior}} = R f^{prior} M \frac{p}{R f^{prior}} \qquad (1)$$

In the above equation, $p = Rf$ (where $Rf$ is the X-ray transformation of the scanned object $f$), which is a non-corrected X-ray projection, where $f$ depends not only on the attenuation coefficient of the material but also on the length of intersection of the rays. The metal projections determined where data in the normalized sinogram were replaced by interpolation. Finally, the corrected sinogram $p^{corr}$ was obtained by the denormalization of the interpolation data and normalized sinogram.

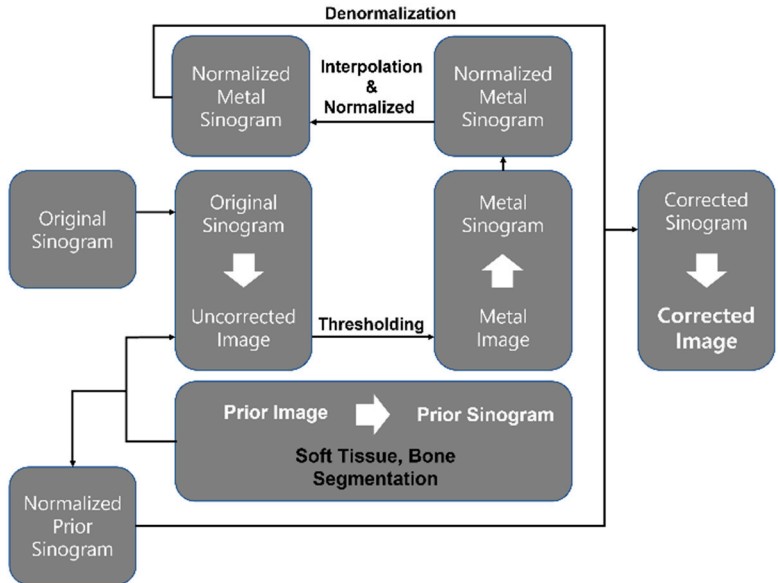

**Figure 1.** Scheme of iMAR. After obtaining a metal sinogram and a prior sinogram from the original sinogram without correction of metal artifacts, the metal sinogram was normalized and interpolated with the original sinogram. Next, the prior sinogram and the uncorrected original sinogram were normalized to form a normalized prior sinogram, and the metal normalized interpolation sinogram and denormalized sinograms were used to generate the final corrected sinogram.

*2.3. Measurement of Metal Artifact*

A total of 10 CT images acquired using the self-made phantom was quantitatively analyzed, including 5 images without total metal artifact reduction and 5 images acquired using iMAR. Since there was no change in the location information of the reconstruction phantom image acquired under each condition, the slice with the maximum number of artifacts shown was selected. Figure 2b shows that the change in artifact signal, according to the distance between the inner and the outer area, was measured.

The signal intensity according to the distance was evaluated as a profile using Image J software (https://imagej.nih.gov/ij/index.html) (accessed on 21 March 2021). The HU value was measured according to the change in distance that was connected on the same line under the same coordinates, as shown in Figure 2b. A paired *t*-test (SPSS 20.0, Chicago, IL, USA) was performed to compare the average of the HU values of the artifacts on each condition CT image, including the iMAR-applied image.

The signal intensity according to the distance was evaluated as a profile using Image J software, and the HU value was measured according to the change in distance that was connected on the same line under the same coordinates, as shown in Figure 2b. The

difference in HU values before and after iMAR application of the images acquired under each condition was analyzed via paired *t*-tests (SPSS 20.0, Chicago, IL, USA).

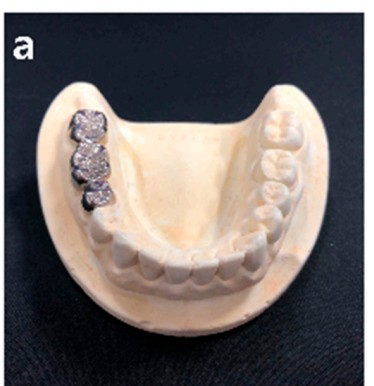 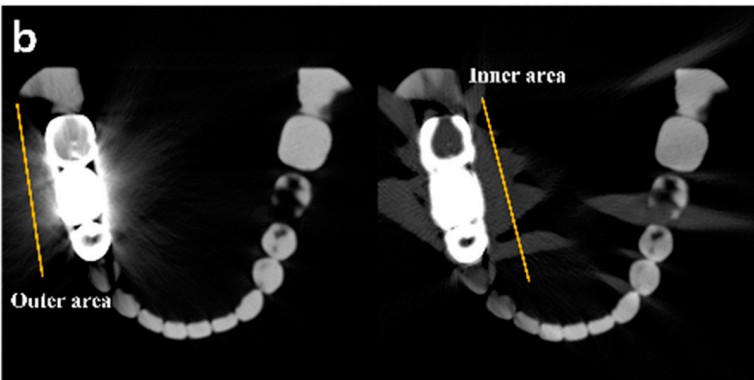

**Figure 2.** Reference phantom composition and metal artifact measurement. (**a**) A prosthesis made of nickel alloy was inserted into the right mandibular area first and second molars and first premolars. (**b**) Image acquired under 120 kVp and 50 mAs. The left image is a non-iMAR image, and the right image was acquired after iMAR. The solid line drawn in the inner and outer areas was used to measure artifacts, as described in the Materials and Methods section.

### 2.4. 3D Printing Processing

To obtain the acquired CT image in 3D, the image from which the beam hardening artifact was removed was first deleted using Xelis 3D (Infinity Healthcare, Seoul, Korea) software. For the 3D image of the phantom, surface data were acquired using InVesalius 3.1.1 software (https://invesalius.github.io) (accessed on 21 April 2021), and surface data were specified by selecting the target area corresponding to the area of interest and excluding unnecessary area. Using Rhino 6 (https://www.rhino3d.com/kr) (accessed on 21 April 2021) for the STL file-converted surface 3D image, unnecessary regions, except for the target area, were precisely removed to eliminate printing restrictions and obtain the optimal image for printing. Therefore, a total of two impressions of teeth before and after applying iMAR in the images acquired at 120 kVp 50 mAs were completed. The printer used was a UV DLP (digital light processing) 3D printer. The 3D printer was able to print up to a stacking thickness of 0.05 mm. Each cross-section CT image was reconstructed by interpolation to 0.05 mm in the Xelis program, and, finally, the 3D printer output to a thickness of 0.05 mm. The material used was a standard photopolymer resin, and the UV LED set exposure time to 1.7 s for the 3D printing. A method in which the output product had little error compared with the actual phantom was used.

### 3. Results

#### 3.1. CT Image Metal Artifact Evaluation

Theoretically, it is known that the higher the tube voltage, the less the occurrence of metal artifacts [17,18]. In this study, the degree of the existing metal artifact generation environment was compared and evaluated according to changes in tube voltage and tube current. The HU values of the inner and outer areas of the image acquired with high tube voltage decreased in the data without iMAR application (Table 1).

In this study, the degree of occurrence of artifacts before/after applying the iMAR technique under similar image acquisition condition was of interest. Figure 3 shows the signal intensity of the artifact according to the changes in image acquisition parameters and iMAR application.

As shown in Figure 3a, the HU value decreased as the tube voltage increased in the image without iMAR application. In addition, the signal change in the artifact, according to the change in tube current, is shown in Figure 3b. Analysis of the HU values of the inner and outer sections before and after iMAR application under the same conditions revealed lower average values in all conditions (Table 1). The change in HU value according to the change

in tube current did not increase significantly in the absence of iMAR in contrast to changes in the tube voltage condition. In this experiment, the effect of a beam hardening artifact was greatly affected by kVp rather than tube current. In addition, the occurrence of artifacts before and after iMAR application under each imaging acquisition parameter is expressed in Tables 1 and 2, and lower HU values were confirmed in all experimental conditions for the images subjected to iMAR. The lowest HU value was obtained at 120 kVp and 80 mAs. In addition, the greatest difference in HU value before and after iMAR application occurred at 80 kVp and 50 mAs.

**Table 1.** Average artifact signal intensity (Hounsfield unit) before and after iMAR application, according to changes in tube voltage.

| | 80 kVp 50 mAs | | 100 kVp 50 mAs | | 120 kVp 50 mAs | |
|---|---|---|---|---|---|---|
| | **Inner Area** | **Outer Area** | **Inner Area** | **Outer Area** | **Inner Area** | **Outer Area** |
| iMAR | −501.20 ± 369.24 | −720.80 ± 631.69 | −434.11 ± 432.70 | −704.18 ± 318.15 | −438.65 ± 344.62 | −575.63 ± 425.25 |
| non iMAR | −76.92 ± 631.69 | −5.099 ± 829.2 | −98.083 ± 457.227 | −120.738 ± 669.435 | −298.416 ± 288.370 | −278.274 ± 461.582 |

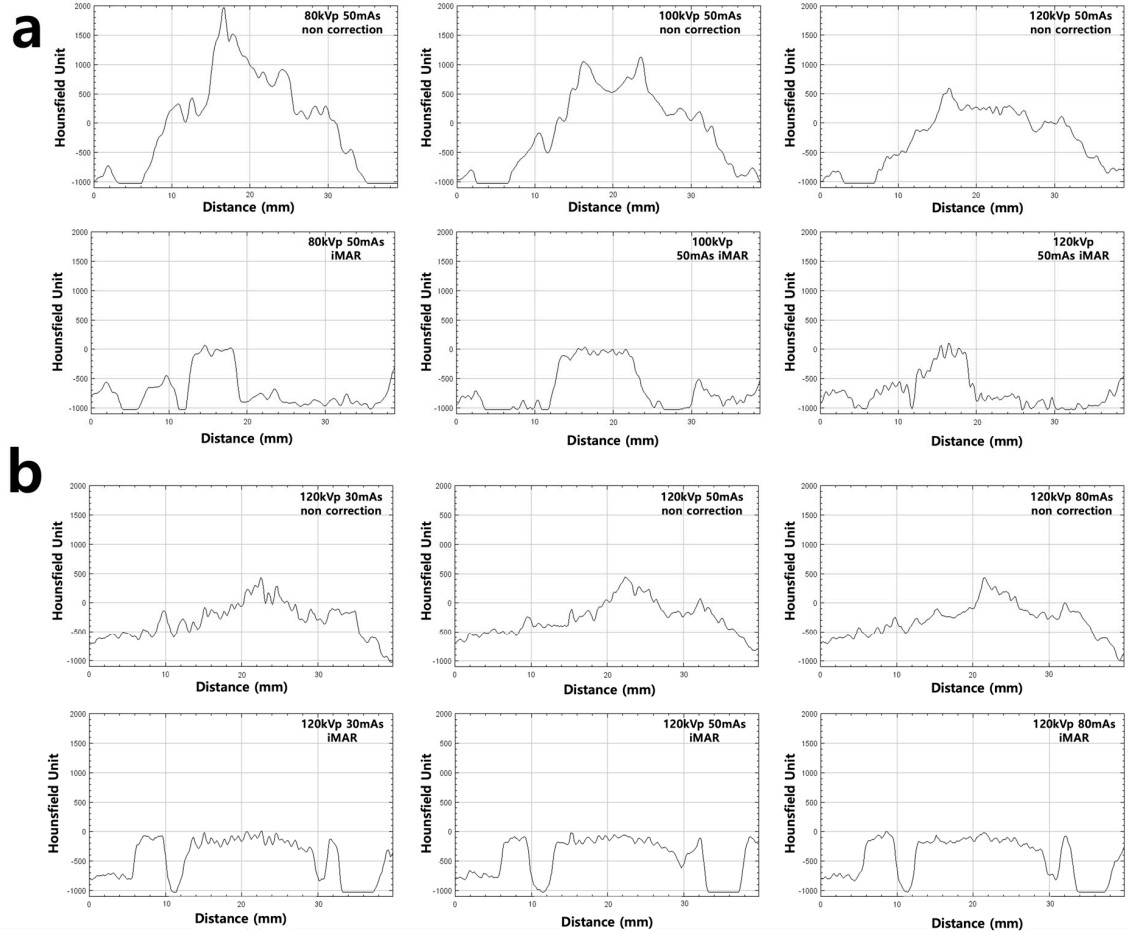

**Figure 3.** The HU change according to the distance of region of interest under each inspection condition before and after iMAR. (**a**) Change in HU value of inner area according to tube voltage change. In the absence of iMAR, the HU value decreased as the tube voltage increased, and following iMAR, the HU did not change significantly. (**b**) Change in HU value of the outer area according to change in tube current.

**Table 2.** Changes in Hounsfield unit values according to change in current after fixing tube voltage in the artifact area.

| | 120 kVp 30 mAs | | 120 kVp 50 mAs | | 120 kVp 80 mAs | |
|---|---|---|---|---|---|---|
| | Inner Area | Outer Area | Inner Area | Outer Area | Inner Area | Outer Area |
| iMAR | −448.98 ± 347.59 | −597.77 ± 411.40 | −438.65 ± 344.62 | −575.63± 425.25 | −438.66 ± 344.70 | −668.73± 374.19 |
| non iMAR | −310.52 ± 298.20 | −271.21 ± 514.26 | −298.42± 288.37 | −278.27 ± 461.58 | −266.78 ± 288.31 | −280.73± 487.60 |

### 3.2. 3D Printing Product

The 3D printing process before and after iMAR application was acquired by converting the CT image into an STL file using InVesalius 3.1.1 software, as shown in Figure 4a. Threshold adjustment for 3D printing was performed, as shown in Figure 4b. Since the artifact was expressed as a signal value in the image, it was possible to adopt a more accurate tooth appearance in the iMAR-applied CT image. Subsequently, as shown in Figure 4b,c, two data impressions were acquired as STL files, respectively, and 3D printing was successfully completed. Figure 4b is a 3D output of a CT image without iMAR application, with the beam hardening artifact in the form of persistent flash output. However, in the case of the output acquired with iMAR (Figure 4c), the artifact was, relatively, not applied. However, due to the application of a threshold to activate the area of the tooth except for the metal artifact, the signal value of the tooth originally expressed was lost, so the shape could not be perfectly expressed. In addition, structurally, in the case of the second premolars extracted via 3D printing, due to the greater influence of artifacts where the metal structure was straight, excessive signal loss was observed, and the output of the corresponding area was associated with excessive loss (Figure 4c). This phenomenon was also observed in the region where iMAR was not applied.

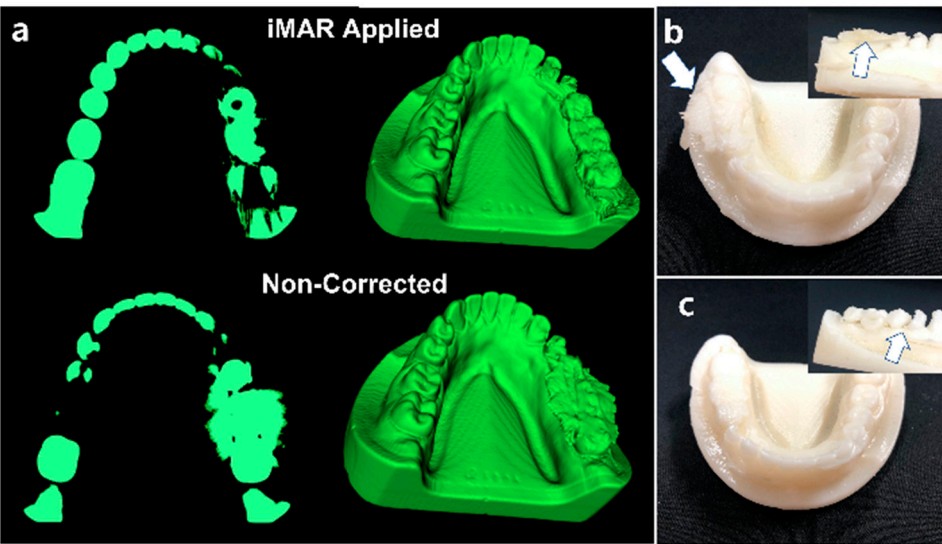

**Figure 4.** 3D printing of CT image. (**a**) The selected area for 3D printing in InVesalius 3.1.1 software. (**b**) As a 3D printing product that did not apply iMAR, the artifact was printed as it was (white arrow). (**c**) A 3D-printed product that was iMAR-applied where relatively large portions of artifacts have been removed. However, it was printed in the form of a defect in the inferior part of the metal area. In addition, in the case of the second premolar, it was generated as a defect due to excessive signal loss (arrow).

### 4. Discussion

In this study, the conditions for reducing the metal artifacts generated by dental prosthetics during dental-impression acquisition were analyzed and the iMAR technique was applied to remove the artifacts to generate accurate output via 3D printing. Recent efforts involved the analysis of the surface of the human body using 3D printing and surgical approaches via specific bone registration and symmetrical implants [19–21]. CT

is a medical device that can accurately delineate human anatomy and phase information. Since CT can accurately determine the exact size and position of the human body, it can accurately implement the model and size, which represent the anatomical information of the image [22,23]. In particular, since CT is expressed in HUs based on the atomic number and density of a substance, it facilitates the extraction of 3D surface information of a specific density for 3D printing protocols [24–26]. However, metal artifacts represent one of the biggest limitations in 3D printing of CT images. A metal artifact is a filter phenomenon in which the low energy of X-rays is absorbed when X-rays pass through a metallic structure, and the average energy of the X-rays is increased [27]. Therefore, it is expressed in the form of a flash centered on the location of the metal area. As for the artifact, the signal increases in the existing anatomical position, that is, the HUs increase so that the artifact in the form of flashes is inevitably generated in the 3D-printed output. The phantom used in this study was not only the human body structure in which metal artifacts are most commonly generated, but also the teeth, which are a relatively sophisticated structure in the human body. Therefore, in this study, teeth with metal components were used as a target to obtain 3D printing impressions.

The method of reducing beam hardening artifacts is commonly used by increasing the tube voltage technically. In this study, as well, it was possible to observe a high artifact reduction effect at high tube voltage during the optimization with reduced artifacts. In the absence of iMAR, the HU values of the inner and outer areas were shown the low as applying a high tube voltage. In the absence of iMAR, the HU values of the inner and outer areas were shown the low as applying a high tube voltage. In the absence of iMAR, the HU values of the inner and outer areas were shown the low as applying a high tube voltage. In the case of images subjected to iMAR, the reduction in metal artifacts at a low tube voltage of 80 kVp was about 200% greater than in iMAR at 120 kVp. Thus, the elimination of artifacts at low tube voltage was higher following iMAR. Therefore, considering the patient's exposure, the use of a low tube voltage and iMAR was a very effective CT scanning method. However, since the purpose of this study was to reduce artifacts to the maximum extent feasible and output them using a 3D printer, data obtained under high voltage were used in this experiment. Data acquired under high voltage suggested a greater effect of normalization because of low HU values in the neighboring tissue in the prior image data from which the metal signal was deleted following iMAR application. When data were acquired at high tube voltage and iMAR, the artifacts were lower. In conclusion, the 3D-printed output was enhanced when data were acquired at high tube voltage and images were subjected to iMAR. Therefore, as shown in Figure 4b, the shape of the metal artifact remained unchanged in the images not subjected to iMAR, so it was not possible to present the shape of the tooth. However, in the case of Figure 4c, metal artifacts were eliminated partially, and the original phantom shape was reproduced.

In this study, the signal value of the structure under the metal component was low, which was observed in the form of a defect in the 3D-printed output, presumably due to a geometric phenomenon during the CT scan of the phantom. In general, during CT, data are collected in a helical form on the same plane as the arrangement of dentition. In this study, data were collected in the vertical direction of the plane of dentition. As for metal artifacts, if there is a metal structure in the X-ray scanning direction during CT, low-energy X-rays are absorbed as the X-rays pass through the metal, resulting in an increase in average energy. Thus, in the presence of a metal structure in the direction of X-ray scan, the average energy in the scanning axial direction reduces the uniformity and affects the HU of the subject. Therefore, if the geometrical position of the metal is considered and the angle of the CT scan is determined, the occurrence of artifacts in the raw data can be adjusted. Further, it is expected that this data loss part would be sufficiently accurate if computer-aided design (CAD) using medical images or specific software is used. In this study, it was meaningful to evaluate the degree of metal artifact reduction according to the application of the iMAR technique and to confirm the possibility of 3D printing. We did not use 3D CAD using commercially available medical images, but reduced artifacts using the iMAR algorithm

with free software and limited the evaluation of its utility in 3D printing tasks. Therefore, in this study, this entailed a direct application of basic knowledge regarding the theory of CT image acquisition and methods used to reduce artifacts that are generated. It was a study that could be directly applied in actual clinical practice. Therefore, it could be said that a method for setting conditions and reducing CT artifacts in patients implanted with metal prostheses and implants was proposed.

This study has several limitations. As mentioned earlier, when 3D printing was performed using iMAR, the beam hardening artifacts could be reduced effectively. However, when software was used to design bone alignment and assistive technology devices based on recent medical images, a more elaborately calibrated printout was feasible. Second, the geometrical accuracy of the 3D printed product and the reference phantom could not be evaluated. For this purpose, a comparative measurement of marginal and internal fit with the reference phantom was needed. There were many factors that caused errors in the geometrical accuracy, but the performance of the 3D printer used was the main factor, suggesting the need for related studies.

### 5. Conclusions

In conclusion, if the iMAR used in this study was applied and data was acquired using a high tube voltage, a significant number of the metal artifacts were removed, so that a 3D-printed output that was almost similar to a human body model was possible.

**Author Contributions:** In addition, all authors were actively involved in the study in different capacities: writing—original draft preparation, S.-M.Y.; writing—review and editing, Y.P.; investigation and measurement visualization, Y.P. and S.-M.Y. All authors have read and agreed to the published version of the manuscript.

**Funding:** This study was supported by the Basic Science Research Program through the National Research Foundation of Korea (NRF), funded by the Ministry of Science, ICT & Future Planning (Grant No. 2018R1D1A1A02085800 and 2021R1F1A1056078).

**Institutional Review Board Statement:** Not applicable.

**Informed Consent Statement:** Not applicable.

**Data Availability Statement:** The data presented in this study are available upon request from the corresponding author.

**Acknowledgments:** This research was conducted with cooperation from Siemens Korea.

**Conflicts of Interest:** The authors declare no conflict of interest.

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
