# Peer review of "3D Printing of Tooth Impressions Based on Multi-Detector Computed Tomography Images Combined with Beam Hardening Artifact Reduction in Metal Structures"

_applsci, doi:10.3390/app12073339_

Round 1
Reviewer 1 Report
- Typographical edits and grammar should be improved.
- Comparative study of this method with other known technique that makes tooth imprint.
Author Response
- Typographical edits and grammar should be improved.
Answer:
The incorrect part of the sentence in the introduction has been improved, and the entire sentence has been reviewed once again. Thank you for helping to make for good research paper.
Revisoin:
“During periodontal diseases such as tooth decay, the shape of the tooth is acquired using the dental impression. The treatment entails insertion of a dental implant. An irreversible hydrocolloid material known as alginate is used to obtain the dental impression conventionally [1, 2]. However, in this conventional method, since the alginate must be firmly maintained on the teeth, the patients may complain of discomfort during the acquisition method.”
- Comparative study of this method with other known technique that makes tooth imprint.
Answer:
Thanks for the good point. This study is a study that performed tooth impression by reducing metal artifacts. As far as we know, it is almost the first study to apply iMAR and perform research using a multi detector. Therefore, the contents have been modified as follows so that these contents can be emphasized.
Revisoin:
- Title : 3D printing of tooth impression based on multi detector computed tomography images combined with beam hardening artifact reduction of metal structures
- 2.1 material and methods : All 64 channel multi detector CT images were performed using a (Somatom singo, Siemens, German) and CT scans based on the changes in five parameters.

Reviewer 2 Report
The study shows the a possible strategy to reduce artifacts on a phantom that presents nickel-based insert by using micro-CT analysis. The study clearly shows the results obtained and possible strategies that can be adopted to reduce this effect. In my opinion, the paper deserves publication in Applied Sciences.
Minor comments to be addressed are reported below:
Line 28 – the text does not start with a capital letter.
Line 128 – use a reference number for ImageJ and put the link in the bibliography.
Lines 141 and 143 – as for ImageJ use reference numbers and put the link of the software in the bibliography.
Figure 3 – the resolution is too low. Numbers are not visible. Increase the resolution and the text of the figures.

Author Response
- Line 28 – the text does not start with a capital letter.
Answer:
Missing sentences that occurred in the process of submitting this thesis have been re-inserted. Thank you for your careful review.
Revision:
- introduction : “During periodontal diseases such as tooth decay, the shape of the tooth is acquired using the dental impression. The treatment entails insertion of a dental implant. An irreversible hydrocolloid material known as alginate is used to obtain the dental impression conventionally [1, 2]. However, in this conventional method, since the alginate must be firmly maintained on the teeth, the patients may complain of discomfort during the acquisition method.” Insertion
- Line 128 – use a reference number for ImageJ and put the link in the bibliography.
Answer:
Thanks for the good point. I have inserted a soft link in the first sentence about Image J software. Also, links to freeware used in this study are inserted in the body of the article. I would appreciate your understanding for not inserting it as a reference so that this thesis can be written in a consistent way.
- Figure 3 – the resolution is too low. Numbers are not visible. Increase the resolution and the text of the figures.
Answer: Thank you for your kind comments for improving the quality of this paper. As the reviewer pointed out, I increased the resolution of figure 3 to 500 dpi. Thank you again.
Revisoin: The resolution of all figures has been increased to 500 dpi.

Reviewer 3 Report
- Lack of iMAR explicit in Abstract.
- Introduction must be verify and rewritten.
- Numerical references are not in agreement with the text (missing references 1, 2) in the Introduction.
- The Materials (information on material type, chemical structure, etc. is missing. ) and 3D printer (2.4) used should be described with sufficient details to allow others to replicate the results.
- All figures and tables should be cited in the main text as Figure 1, Table 1, etc.
- The position of the figures must be revised (as close as possible to the place of their description, see Figure 3 as an example), the equations must be numbered (see template), the notation of the tables must be checked (Table 1.1. In text Table 1).
- Line 176 Tables 1 and 2?
- Is the research design must be reconsidered.
- The results not clearly presented.
- References must be in accordance with the settings required by the template.
Author Response
- Lack of iMAR explicit in Abstract.
Answer:
When using the abbreviation of iMAR, which is the first expression in Abstract, an ‘iterative metal artifact reduction algorithm’ is inserted so that the meaning can be accurately conveyed. Thank you for your careful review.Revision: iterative metal artifact reduction algorithm(iMAR) insert.
- Introduction must be verify and rewritten.
Numerical references are not in agreement with the text (missing references 1, 2) in the Introduction.
Answer: Answer:.
Thanks again for the good point. This journal has been rewritten by correcting the omissions that occurred during formatting the journal. Also, the missing parts of references 1 and 2 have been corrected again. Thank you again for your careful review.
Revision:
During periodontal diseases such as tooth decay, the shape of the tooth is acquired using the dental impression. The treatment entails insertion of a dental implant. An irreversible hydrocolloid material known as alginate is used to obtain the dental impression conventionally [1, 2].
- The Materials (information on material type, chemical structure, etc. is missing. ) and 3D printer (2.4) used should be described with sufficient details to allow others to replicate the results.
Answer:
Through this paper, the detailed contents of 3D printing have been rewritten along with the revised contents so that other researchers can conduct reproducibility experiments and studies. Thank you for helping to complete a good research thesis.
Revision:
Each cross-section CT image was reconstructed by interpolation to 0.05mm in the Xelis program, and finally, 3D print output to a thickness of 0.05mm. The material used is a standard photopolymer resin, and the UV LED set exposure time at 1.7 sec for 3D printing.
- All figures and tables should be cited in the main text as Figure 1, Table 1, etc.
The position of the figures must be revised (as close as possible to the place of their description, see Figure 3 as an example), the equations must be numbered (see template), the notation of the tables must be checked (Table 1.1. In text Table 1).
Answer:
As the reviewer pointed out, in the course of the development of the content of the thesis, it was decided that it was right to change the order of figure 1 and figure 2, so it was readjusted. Also, the order of the figures was reversed, so the position of the figures in the paper was readjusted. The order of all figures has been adjusted by changing the order of the figure citation after the sentence or related content. Thanks again for helping to make this a good thesis.
Revision:
- Reposition figure 1 and figure 2 for connection to the context of the paper
- Reposition by changing the order of Figure 1 to section 2.2.
- Reposition by changing Figure 2 to 2.3section
- Reposition by changing Table 1 to section 3.1
- Reposition by changing the figure 3 to section 3.1
- Reposition figure 4 to the last paragraph in section 3.2
- Line 176 Tables 1 and 2?
Answer:
Table 2 was re-inserted because the contents of Table 2 were omitted during the editing of the thesis. Thank you again for your careful review.
Revision: table 2 insertion
- Is the research design must be reconsidered.
Answer:
Specific details about 3D printing have been added to the materials and methods section. In addition, the contents that the reviewer pointed out were reinforced along with the composition of the thesis and the parts that were not well understood. This study performed tooth impressions by reducing metal artifacts. As far as we know, it is almost the first study to apply iMAR and conduct research using a multi-detector. The study of acquiring an impression of the human body can be carried out by overcoming the limitations through this study by continuously applying the iMAR-applied research. I cordially ask you to understand this part. To emphasize this part, along with the title, the point of using the multi-detector in the CT equipment of the material method part has been corrected. We ask that the reviewer review positively so that experiments using multi-detector can proceed in the future.
Revision: Title : 3D printing of tooth impression based on multi detector computed tomography images combined with beam hardening artifact reduction of metal structures
- The results not clearly presented.
Answer:
In this study, by applying iMAR by artificially inducing a metal artifact, the signal intensity generated by the artifact was measured, and the difference was identified. The conditions for optimal 3D printing are presented in the results. The results of 3D printing performed under these optimal conditions are presented through figures. As you pointed out, there is a limitation in not giving the quality of the output through 3D printing as a result. This part was presented as a limitation of this study in the limitation part of the discussion part. Although there are many shortcomings, we would like a positive review from the reviewer so that more advanced research can be carried out in the future as an initial study of 3D printing applied with iMAR. In addition, The conclusion part of this experiment has been summarized in one sentence. again so that the results and conclusions can be made more apparent to the reader. Thanks again for your careful review and accurate comments.
Revision:
In conclusion, if the iMAR used in this study is applied and data acquired using a high tube voltage, a significant number of the metal artifacts are removed, so that a 3D printed output that is almost similar to a human body model is possible.
- References must be in accordance with the settings required by the template.
Answer:
I checked the reference style and template of this journal and rechecked it as a whole so that it could be corrected correctly. Once again, thank you for the careful review to make this a good research thesis.

Round 2
Reviewer 1 Report
English/sentence formatting should be revised thoroughly prior to accepting the manuscript.
Grammatical statements should be more eloquently presented and avoid long sentences.
Author Response
We thank you for the careful review of this paper. We tried to keep the sentences of the thesis as concise as possible. We also commissioned a professional reviewer to proofread our thesis in English. Finally, a confirmation of English proofreading of this thesis was also issued. Once again, we thank you for the careful review, and we sincerely thank you for helping us complete the final research paper.
Reviewer 3 Report
There are still some problems in writing the figures in the text (see line 123, 129, ...), also the same observation for the Tables (see line 165, ..).
Author Response
Thank you for pointing out the parts that do not fit the context of the sentence. We have corrected that sentence. Once again, thank you very much for your careful review.